# Promoter Methylation Changes in *KRT17*: A Novel Epigenetic Marker for Wool Production in Angora Rabbit

**DOI:** 10.3390/ijms23116077

**Published:** 2022-05-28

**Authors:** Yang Chen, Zhiyuan Bao, Ming Liu, Jiali Li, Yingying Dai, Fan Wang, Xiyu Zhang, Pin Zhai, Bohao Zhao, Xinsheng Wu

**Affiliations:** 1College of Animal Science and Technology, Yangzhou University, Yangzhou 225009, China; yangc@yzu.edu.cn (Y.C.); dx120210143@stu.yzu.edu.cn (Z.B.); mliu1994@163.com (M.L.); dx120200140@stu.yzu.edu.cn (J.L.); mx120200809@stu.yzu.edu.cn (Y.D.); mz120201366@stu.yzu.edu.cn (F.W.); mz120201361@stu.yzu.edu.cn (X.Z.); bhzhao@yzu.edu.cn (B.Z.); 2Institute of Animal Science, Jiangsu Academy of Agricultural Sciences, Nanjing 210014, China; zhaipin@163.com

**Keywords:** *KRT17*, epigenetic marker, wool production, SP1

## Abstract

Wool production is an important economic trait of Angora rabbits. Exploring molecular markers related to wool production is one of the essentials of Angora rabbits’ breeding. *KRT17* (*Keratin 17*) is an important gene of hair follicle development, which must be explored for genetic/epigenetic variation to assess its effect on wool production. Based on the effective wool production data of 217 Angora rabbits, the high and low yield groups were screened with 1.5 standard deviations of the population mean. The full-length sequence of *KRT17* was obtained by rapid amplification of cDNA ends technology, and the polymorphism was analyzed in the promoter, exon, and intron regions by direct sequencing. *KRT17*, *SP1* over-expression plasmids, and siRNA were constructed and transfected into dermal papilla cells. The mRNA expressions of relevant genes were analyzed by RT-qPCR. The methylation level of the KRT17 promoter was determined by Bisulfite Sequencing PCR. Dual-luciferase system, site-directed mutagenesis, and electrophoretic mobility shift assays were used to analyze the binding relationship between SP1 and the promoter of *KRT17.* The structure map of *KRT17* was drawn, and no SNPs were found in the promoter, exon, and intron, indicating a relatively conserved structure of *KRT17*. Expression of *KRT17* was significantly higher in cutaneous tissues than in other tissues and was significantly upregulated in the high-yield group compared to the low-yield group (*p* < 0.05). Furthermore, the overall high methylation levels of *KRT17* CpG I and CpG III showed significant association with low wool yield; the methylation levels of 5 CpG locus (CpG I site 4 and CpG III site 2–5) were significantly different between the high and low yield groups (*p* < 0.05). The methylation levels of 3 CpG locus (CpG I site 4 and CpG III site 4, 14) showed a significant correlation with *KRT17* expression (*p* < 0.05). Overall, CpG III site 4 significantly affects wool production and *KRT17* expressions (*p* < 0.05). This site promotes SP1 binding to the *KRT17* promoter region (CGCTACGCCC) to positively regulate the *KRT17* expression. *KRT17* CpG III site 4 can be used as candidate epigenetic markers for the breeding of high wool-producing Angora rabbits.

## 1. Introduction

Rabbit wool is a high-grade textile raw material of high economic value [1]. Studying the genetic/epigenetic variation of key wool-producing genes can identify molecular markers related to wool production in Angora rabbits. SNP markers have been widely used to assist breeding. For instance, SNPs related to dorsal hair development, such as *FGF5* [2,3], *FGFR2* [4], and *CYP19A1* [5], have been screened. In this study, we mined the SNPs of the *Keratin 17* (*KRT17*) gene in the Angora rabbit population. *KRT17* belongs to the type I keratin genes and is involved in hair follicle development and hair growth. Notably, the *KRT17*-knockout mice exhibited severe hair loss during the first week of life suggesting *KRT17* role in hair fragility and apoptosis [6]. *KRT17* interacts with TRADD via TNFα/TNFR1 signaling pathway to affect the hair follicle growth cycle [7]. *KRT17* also interacts with other keratin family genes to jointly control hair growth [8]. The promoter, exon, and intron sequences of *KRT17* in Angora rabbits have been studied but no corresponding SNPs sites were found as molecular markers. Hence, the epigenetic markers of *KRT17* must be examined.

Currently, there are few studies on DNA methylation in rabbits that mostly focused on identifying the whole-genome DNA or related gene methylation level. For example, a study showed early gene demethylation and re-methylation in cloned rabbit embryos than in normal rabbits [9]. The methyl-CpG-binding protein CIBZ inhibits rabbit myogenic differentiation by directly inhibiting the expression of myogenin [10]. Whether *KRT17* expression is regulated by DNA methylation and if that affects wool production remain unknown. Additionally, *KRT17* expression is regulated by various molecules. Cytokine IL-7A along with transcription factors (TFs) STAT1 and STAT3 induce *KRT17* expression in keratinocytes; similarly, IL-2 and IL-6 are also known to regulate *KRT17* expression [11]. The *KRT17* promoter has multiple TF binding sites (such as including SP1, AP2, and NF1) that contain CpG sites [12]. Whether the regulation of DNA methylation of *KRT17* can regulate its functions affecting the binding of TFs needs to be investigated.

It was shown that *KRT17* is involved in the cyclical development of hair follicles suggesting its role in the wool-producing performance of Angora rabbits [13]. Accordingly, this study examined the relationship between the genetic variation and DNA methylation pattern of *KRT17* to understand its impact on the wool-producing performance of Angora rabbits. The gene expression regulation was investigated, and SNP and DNA methylation sites with significant influences on wool-producing performance were screened out. These may be valuable genetic markers for the molecular breeding of Angora rabbits.

## 2. Results

### 2.1. Screening and Grouping of High- and Low-Yield Angora Rabbits

The production data of 217 Angora rabbits were recorded (Table 1). The weight and wool production of the whole population showed normal distribution (Figure 1). The high- and low-yield group were screened out based on the standard mean deviation of ±1.5 (Mean ± 1.5 SD) (Table 2). The weight and wool production of the high-yield group were 4078.0 ± 200.2 and 443.2 ± 20.8 g. The same were 3887.1 ± 190.3 and 236.2 ± 3.6 g for the low yield group, respectively. Compared with the colony, weight showed no correlation between the high-and low-yield groups (*p* > 0.05); however, there was a significant difference between the wool production (*p* < 0.05). This indicated that change in wool production was not affected by weight change.

### 2.2. Highly Conserved KRT17 Sequence

The 1608 bp full-length sequence of *KRT17* was obtained by RACE assay, including 72 bp 5′UTR, 1299 bp open reading frame (ORF), and 237 bp 3′UTR (with PolyA tail) (Figure 2). The structural diagram of the gene in Figure 3 shows that *KRT17* contains eight exons and seven introns, belonging to the type I keratin gene group. The eight exons, seven introns, and promoter sequences of *KRT17* were amplified from 217 individuals by PCR and analyzed by gel electrophoresis indicating the successful amplification of corresponding target fragments (Figure 4). In addition, no mutation sites were found. This suggested that *KRT17* sequences are highly conserved.

### 2.3. KRT17 Expression Is Closely Related to Hair Follicle Development in Angora Rabbits

Tissue expression profiling showed that *KRT17* expression in cutaneous tissues was significantly higher than in other tissues (*p* < 0.01). The expression was relatively high in the lung and small intestine, and almost absent in the heart, muscle, and brain (Figure 5A). We used the Angora rabbit hair follicle periodic regeneration model to detect the expression of *KRT17* during hair follicle development. The expression gradually increased from 0–150 d reaching the highest at 150 d (*p* < 0.05). Then, the expression began to decrease returning to 30 d level after 180 d, forming a cycle (Figure 5B). This suggested the *KRT17* role in hair growth. Moreover, *KRT17* expression was significantly higher in the high yield group than in the low yield group (*p* < 0.05) (Figure 5C), suggesting that upregulated *KRT17* impacted the wool-producing performance of rabbits.

Next, *KRT17* was knocked down in rabbit dermal papilla cells, and the results showed that WNT2 and FGF2 were downregulated showing a significant difference (*p* < 0.05), while SFRP2 and BMP2 were significantly upregulated (*p* < 0.05) (Figure 5F,G). The upregulation of *KRT17* upregulated WNT2 and FGF2 but downregulated TGFβ1, SFRP2, and BMP2, all showing extremely significant differences (*p* < 0.01) (Figure 5D,E). These results demonstrated that *KRT17* expression played a key role in hair follicle development by altering the expressions of related genes.

### 2.4. Methylation Level of CpG Island in KRT17 Promoter

There were three CpG islands found in the promoter region of *KRT17*. Among them, CpG I is located −1567 to −1403 bp (length 165 bp); a total of 16 CpG sites were detected. CpG II is located −1186 to −1053 bp (length 134 bp); a total of 10 CpG sites were detected. CpG III is located −97 to +120 bp (length 218 bp); a total of 18 CpG sites were detected (Figure 6A). The overall methylation level of CpG I-III in high and low yield groups was analyzed by BSP sequencing (Figure 6B–D). It was found that CpG I methylation was 45.20 and 63.75% in high and low yield groups, respectively, showing a significant difference (*p* < 0.05). CpG II was 9.00 and 11.00% in high and low yield groups, respectively, showing no significant difference (*p* > 0.05). Due to specific design requirements of BSP primer, CpG III could only be amplified for the fragments containing the last 17 CpG sites. Therefore, only these were used to calculate the overall methylation level, which was 50.00 and 62.16% in high and low yield groups, respectively, showing a significant difference (*p* < 0.05).

### 2.5. Detections of Methylation Levels of Different Sites in KRT17 CpG I–III

The black and white dot patterns of CpG I–III sites in the two hair-yielding groups showed that both CpG I and III sites had different degrees of methylation (Figure 7A,C). CpG II sites showed lower methylation levels, while some did not show methylation (Figure 7B). Furthermore, the methylation rate of each site was calculated. The results showed that, except for a few sites, the methylation levels of CpG I and III sites were higher in the low yield group than in the high yield group, suggesting a strong correlation between high methylation levels of CpG I and III with low wool production (Figure 7D,F). Overall, the methylation rate of all CpG II sites was low and varied in the high and low yield groups in an irregular manner (Figure 7E). Among the 16 CpG sites in CpG I, the methylation of CpG 4 (−1539 bp) significantly correlated with wool production (*p* < 0.05), and the overall low methylation of CpG island significantly correlated with high wool production (*p* < 0.01). There was no correlation between the methylation level of other sites and wool production (*p* > 0.05) (Table 3). There was no correlation between the methylation level of 10 CpG sites or the overall methylation level of CpG II and wool production (*p* > 0.05) (Table 4). Among the 17 CpG III sites, methylation levels of CpG 2, 3, 4, and 5 (−53 bp, −51 bp, −38 bp, −31 bp) were significantly associated with wool production (*p* = 0.037, 0.018, 0.035, 0.037, respectively); the low methylation level of CpG III showed significant correlation with high wool production (*p* < 0.01) (Table 5).

### 2.6. KRT17 CpG III Site 4 Can Be a Candidate Epigenetic Marker

Since the methylation level of CpG II sites was relatively low with no linear correlation with gene expression, only the correlation between the methylation level of CpG I and III and *KRT17* expression was investigated. As shown in Table 6, methylation levels of the CpG I sites (except CpG 1, 2, 6, and 7) negatively correlated with *KRT17* expression. The overall methylation level of CpG I negatively correlated with *KRT17* expression (r = −0.319, *p* = 0.368), in which the methylation level of CpG 10 showed the most significant difference (r = −0.670, *p* = 0.034).

As shown in Table 7, the methylation level of CpG III sites (except CpG 1, 9, 11, and 16) negatively correlated with *KRT17*expression; the overall methylation level of CpG III was negatively correlated with *KRT17*expression (r = −0.442, *p* = 0.200). Specifically, the methylation levels of CpG 4 and CpG 14 were significantly different downregulating *KRT17* expression (r = −0.741, *p* = 0.014; r = −0.728, *p* = 0.017).

Considering the association analysis between the methylation level of *KRT17* CpG I-III sites, wool production, and mRNA expression of *KRT17*, CpG III site 4 was selected as candidate epigenetic markers for wool production in Angora rabbit.

### 2.7. Regulation of KRT17 Promoter Activity by SP1

It was predicted that the three CpG islands contain multiple TF binding sites such as for the TFs SP1, AP-2alph, and NF-1 (Figure 8A). The sequence location of CpG4 in CpG III is the binding site for SP1, suggesting that its methylation might regulate *KRT17* expression by affecting the binding of SP1. pcDNA3.1-SP1 or siRNA-SP1 were constructed and transfected into dermal papilla cells (Figure 8B,C), and then *KRT17* expression was detected for over-expression/knockdown of SP1. The results showed that SP1 over-expression significantly upregulated the expression of *KRT17* (*p* < 0.01), while its knockdown did the opposite *(p* < 0.05). This suggested that SP1 regulates the expression of *KRT17* (Figure 8D,E).

The *KRT17* promoter region deletion expression vectors pGL3-P1-P6 (Figure 9A,B) were constructed for luciferase assay. As shown in Figure 9C, the *KRT17* promoter has multiple active regions (*p* < 0.01), including −1395 to −1094, −1094 to −794, −794 to −470, and −191 to −1 bp, among which CpG4 in CpG III is at −191 to −1 bp. To confirm the regulatory effect of SP1 on the *KRT17* promoter, a site-directed mutant (C to A) was created. It was found that, compared with the wild type, the luciferase activity of the mutant was significantly lower (*p* < 0.01), indicating that the mutation site affected the activity of the *KRT17* promoter. To further verify the SP1 mediated regulation of the *KRT17* promoter, the wild-type or mutant vectors were co-transfected with an SP1 over-expression vector, respectively. It was found that after co-transfection with SP1 significantly increased the wild-type promoter activity (*p* < 0.01), while the promoter activity of mutant transfected cells showed no significant difference (*p* > 0.05). This suggested that SP1 overexpression upregulated the activity of the *KRT17* promoter (Figure 9D,E).

Next, we performed an EMSA test to verify SP1 binding to the *KRT17* promoter region. As shown in Figure 9F, no band appeared in lane 1, indicating a good probe. In lane 2, a complex band formed due to the binding of nuclear protein and biotin-tagged probe, demonstrating SP1 binding to the *KRT17* promoter region. To further verify the specificity of SP1 binding to this sequence, a competitive EMSA experiment was performed (Figure 9G). A new band in lane 2 suggested probe binding to the nuclear protein. The band in lane 3 was lighter, and there was no band in lane 4 due to the competitive binding of the unlabeled probe to the nuclear protein compared to the labeled probe. New bands in lanes 5 and 6 indicated that the normal probe mutation did not affect the binding of the labeled probe. The above results indicated that SP1 specifically binds to the *KRT17* promoter sequence.

## 3. Discussion

In this study, the wool production of the same batch of Angora rabbits varied among individuals, in a range of 259.4 g, indicating that breeding could be improved for wool-producing performance. We previously showed that the gene *KRT17* plays an important role in hair follicle development in rabbits [7,13,14]. Therefore, we speculated whether *KRT17* has an SNP site that could be used as a molecular genetic marker of wool-producing traits in Angora rabbits. However, we found that the structure of *KRT17* is conserved in the population of Angora rabbits showing no SNPs. Therefore, further association analysis with wool production was not performed.

Next, we analyzed the expression of *KRT17*. Using a rabbit hair follicle development synchronization model, we found that *KRT17* expression exhibited periodicity with hair follicle development, and *KRT17* regulated WNT signaling pathway genes, such as *WNT2* [15], *FGF2* [16], *TGFβ* [17], *SFRP2* [18,19], *BMP2* [20], etc. This suggested that *KRT17* could affect the transcription of hair follicle development-related genes playing a positive regulatory role in inducing the transition of hair follicles from telogen to anagen. Moreover, *KRT17* expression in the high yield group was significantly higher than in the low yield group. DNA methylation, one of the epigenetic mechanisms, can cause gene silencing without genetic polymorphisms [21]. Therefore, the effect of *KRT17* promoter DNA methylation on *KRT17* expression was investigated regarding wool-producing performance in Angora rabbits.

As a very important epigenetic change, DNA methylation regulates immune response [22], muscle development [23], stem cell differentiation [24], and many other aspects of animals. Rabbit *KRT17* promoter region CpG islands showed different degrees of methylation in the high and low yield groups. The overall methylation level was lower in the high yield group than in the low yield group, indicating the impact of *KRT17* methylation on wool-producing traits. Furthermore, correlation analysis showed that overall low methylation levels of CpG I and CpG III were significantly associated with high wool production. The methylation levels of the CpG 1 site in CpG I and CpG 4 sites in CpG III were significantly correlated with wool-producing performance. These 5 CpG sites can be used as potential epigenetic markers for wool-producing traits in Angora rabbits.

In general, the gene expression decreases with the increase of methylation level [25]. The overall methylation levels of *KRT17* promoter CpG I and CpG III showed a negative correlation with *KRT17* expression. The 10th CpG site in CpG I showed a significant negative correlation with *KRT17* expression, while the 4th and 14th CpG sites in CpG III also showed a significant negative correlation. This indicated that these three sites might be the key sites regulating *KRT17* transcription. Comprehensive analysis showed that CpG4 in CpG III significantly correlated with both wool production and *KRT17* expression. Therefore, we speculated that this site could be a key differential methylation site and can be used as candidate epigenetic markers for wool-producing performance in Angora rabbits.

Furthermore, the CpG4 CpG III site was tested for specific binding of SP1 with the *KRT17* promoter. SP1 is a member of the SP/KLFs family, which specifically binds to the GC-rich promoter sequences to regulate the transcription of target genes [26]. SP1 is an indispensable transcription factor for biological activities [27,28]. This study found that SP1 expression positively correlates with *KRT17* expression. Through the dual-luciferase detection system, site-directed mutation, and EMSA analyses, we found that SP1 specifically binds to the *KRT17* promoter region. However, DNA methylation inhibiting the binding of SP1 to the *KRT17* promoter demands further detailed studies.

## 4. Materials and Methods

### 4.1. Ethics Statement

All experimental protocols were approved by the Animal Care and Use Committee of Yangzhou University (2020-DKXY-15).

### 4.2. Sample Collection

In addition, 217 adult Wanxi Angora rabbits were raised in the same batch under the same feeding conditions. Angora rabbits were uniformly sheared and, after a 73-d growing period, the gross weight was recorded. Individuals with wool production higher than the population mean by 1.5 standard deviations (Mean + 1.5 SD) were defined as high-yielding individuals, while those lower than this criterion were defined as low-yielding individuals. Individuals with significantly different weights were eliminated to find the final groups of high- and low-yield individuals. Finally, 20 and 11 animals formed the high- and low-yielding groups, respectively.

Rabbit ear tissues were collected from all 217 animals for DNA extraction to detect polymorphisms. Three rabbits from the total were randomly selected, slaughtered after anesthesia, and 10 tissues including skin, heart, liver, spleen, lung, kidney, small intestine, muscle, brain, and stomach were collected for tissue expression profiling. Five rabbits were randomly selected from each group; two copies of the back tissue at the same position were removed, one for fluorescence quantitative analysis and one for methylation detection. Synchronization model samples of the rabbit hair follicle with hair plucking treatment were constructed and preserved in our laboratory in the early stage, including 3 cutaneous tissues at 30, 60, 90, 120, 150, and 180 d each. These hair follicles corresponded to the growth phase 0–120 d, catagen and telogen phase 120–150 d, and a new round of growth phase 150–180 d [15].

### 4.3. Race and Cloning of KRT17 Gene

Three 5′ and two 3′ RACE-specific primers were designed according to the rabbit *KRT17* sequences available on NCBI (Accession No. XM_002719383.3), following the instructions of the Race kit (Invitrogen & Clontech, Carlsbad, CA, USA) primers (Table 8). Then, the full-length cDNA sequence of *KRT17* was assembled.

### 4.4. Polymorphism Detection

The ear tissue DNA was extracted using the tissue genomic DNA extraction kit (DP304) (Tiangen, Beijing, China). The primers were designed to amplify the gene sequence of *KRT17* based on the cross-overlap principle. In total, 17 pairs of primers were used for amplification, including 8 pairs for the exon sequence, 6 pairs for the intron sequence, and 3 pairs for the promoter sequence of the *KRT17* gene (Table 9).

### 4.5. Fluorescence Quantitative Analysis

Real-time PCR was carried out using the ChamQTM SYBR^®^ qPCR Master Mix (Vazyme, Nanjing, China) on an ABI QuantStudio^®^ 5 PCR System with the following program: 1 cycle at 95 °C for 30 s, followed by 40 cycles of 95 °C for 10 s, and 60 °C for 30 s. The fluorescence quantitative primers are listed in Table 10. Each sample was measured in triplicate, and the results were normalized to the *GAPDH* gene. The relative expression of the target gene was calculated by the 2^−ΔΔCt^ method.

### 4.6. Over-Expression Vector Construction and RNAi Interference

The *KRT17* cDNA was reconstructed into the pcDNA3.1 (+) vector with restriction enzymes *Nhe* I and *EcoR* I. Based on the rabbit *KRT17* mRNA sequence obtained from the RACE test results, siRNA and negative control were designed. The siRNA sequence was synthesized by Zimmer Gene (Shanghai, China). In addition, according to the *SP1* sequence (accession number XM_002711128.3), pcDNA3.1^(+)^-SP1 over-expression vector was constructed, and siRNA-SP1 was synthesized, as shown in Table 11.

### 4.7. Cell Culture and Transfection

Rabbit dermal papilla cells were isolated by two-step enzyme digestion and incubated in a DMEM medium at 37 °C and 5% CO_2_. Transfection was performed at 70–80% cell confluency using Lipofectamine 2000 as per the kit instructions.

### 4.8. Bisulfite Sequencing PCR (BSP)

Genomic DNA was modified and purified using an EpiTect Fast DNA bisulfite kit (Qiagen, Hilden, Germany). BSP primers were designed using Meth-Primer software (http://www.urogene.org/methprimer, 10 March 2020) (Table 12). In addition, 100 ng of bisulfite-treated DNA were used in a 50 μL PCR reaction mixture. The reaction program was as follows: 98 °C for 10 s, 55 °C for 30 s, 72 °C for 30 s. After repeating 40 times, the product was kept at 4 °C. Finally, PCR products were cloned into a pMD19-T vector (Takara, Dalian, China), and 30 clones were sequenced for each sample.

The sequencing results were analyzed by BiQ Analyzer software, and the black and white dot plots were made by the online software MSR calculate (http://www.msrcall.com/MSRcalculate.aspx, accessed on 10 March 2020). The TF binding sites on KRT17 CpG island were predicted using the online software AliBaba2.1 (http://gene-regulation.com/pub/programs/alibaba2/index.html, accessed on 10 March 2020).

### 4.9. Luciferase Reporter Assay

The upstream primers of different promoter fragments (P1–P6) of the *KRT17* gene were designed; the downstream primer was the same primer *KRT17*-P (Table 13). These promoter fragments were inserted into the pGL3-Basic vector at *Nhe* I and *Xho I* restriction sites. RAB-9 cells (ATCC) were co-transfected with the pRL-TK vector. The co-transfection of pGL3-Basic empty vector and pRL-TK vector was used as a negative control, and the co-transfection of pGL3-Control vector and pRL-TK vector was used as a positive control. Three experiments were repeated for each group. The transfected cells were collected and analyzed using the Dual-Luciferase Reporter Assay System (Promega, Madison, WI, USA).

### 4.10. Site-Directed Mutagenesis

The P6 vector (pGL3-WT) was used as the wild-type template and primers were designed to insert C to A mutation in the CpG-SP1 binding site CTACGCCC (Table 14). The mutant vector (pGL3-MUT) was constructed similar to the P6 vector.

### 4.11. Electrophoresis Mobility Shift Assay (EMSA)

Cells were pretreated by EMSA assay cell pretreatment kit (Viagene Biotech, Changzhou, China). The nuclear protein was extracted, and the protein concentration was determined by the BCA method. According to the binding site sequence of the TF SP1, the EMSA binding reaction probes were designed, and the 5′-end was labeled with biotin (Table 15). The EMSA binding reaction system is shown in Table 16. The cold competitive reaction system was configured according to Table 17. The samples were analyzed by non-denaturing polyacrylamide gel electrophoresis, transferred, and UV cross-linked before carrying out the chemiluminescence reaction, development, and imaging.

### 4.12. Data Statistics and Processing

The data were statistically analyzed by Excel and SPSS programs. The *t*-test was used to analyze the significance of the difference between the overall methylation rate in the two groups and the methylation rate of a single CpG dinucleotide site. Pearson correlation analysis (PCoA) of methylation levels of CpG island sites and gene expressions was performed. The correlation analysis of methylation levels of CpG I-III sites and wool production was performed using the Fisher test. The 2^−ΔΔct^ method was used to quantify the fluorescence results and the *t*-test was used to analyze the significant differences. Differences in luciferase activity were analyzed by one-way ANOVA.

## 5. Conclusions

*KRT17* sequence is relatively conserved. Its expression in cutaneous tissues was significantly higher than in other tissues, higher in hair follicle development than in other periods, and higher in the high yield group than in the low yield group. It might affect the expressions of downstream genes related to hair follicle development (e.g., *WNT2*, *FGF2*), suggesting a *KRT17* close relation to hair follicle development. Furthermore, *KRT17* CpG III site 4 showed a significant correlation with both wool production and *KRT17* expression and can be used as candidate epigenetic markers for the wool production traits in Wanxi Angora rabbits. This site is the location of SP1 binding that might positively affect the expression of *KRT17*.

## Figures and Tables

**Figure 1 ijms-23-06077-f001:**
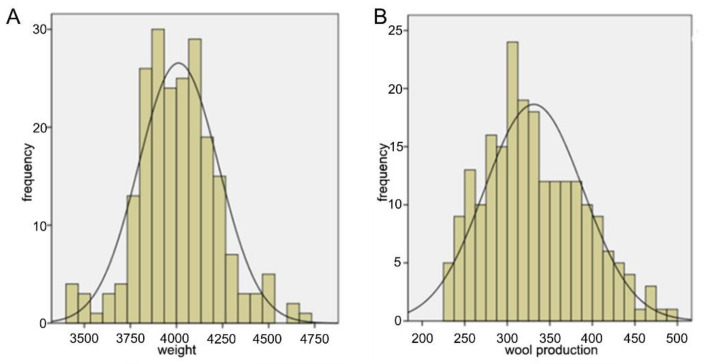
Histograms showing the (**A**) weight and (**B**) wool production data of Angora rabbits.

**Figure 2 ijms-23-06077-f002:**
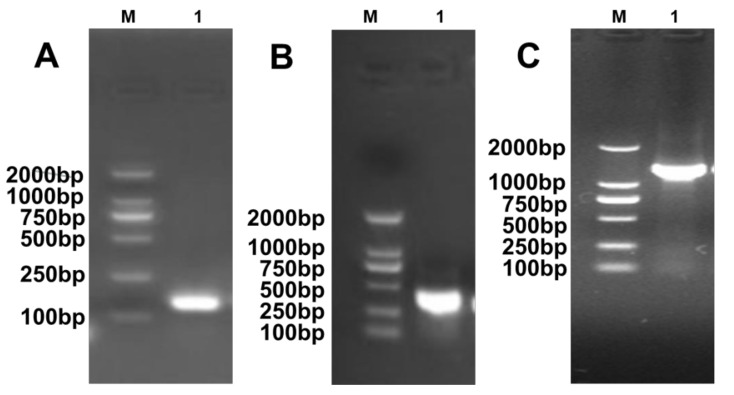
RACE analysis of KRT17 sequence (**A**) 5′UTR; (**B**) 3′UTR; and (**C**) CDS. M refers to the DL2000 Marker.

**Figure 3 ijms-23-06077-f003:**
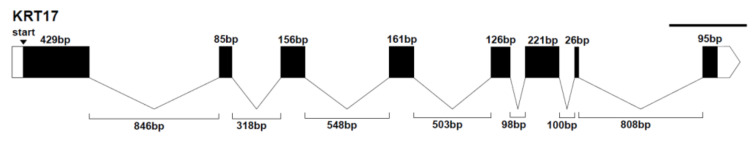
Structure of KRT17. White boxes represent the non-coding regions, black boxes represent the exons, and the broken lines represent the intron regions.

**Figure 4 ijms-23-06077-f004:**
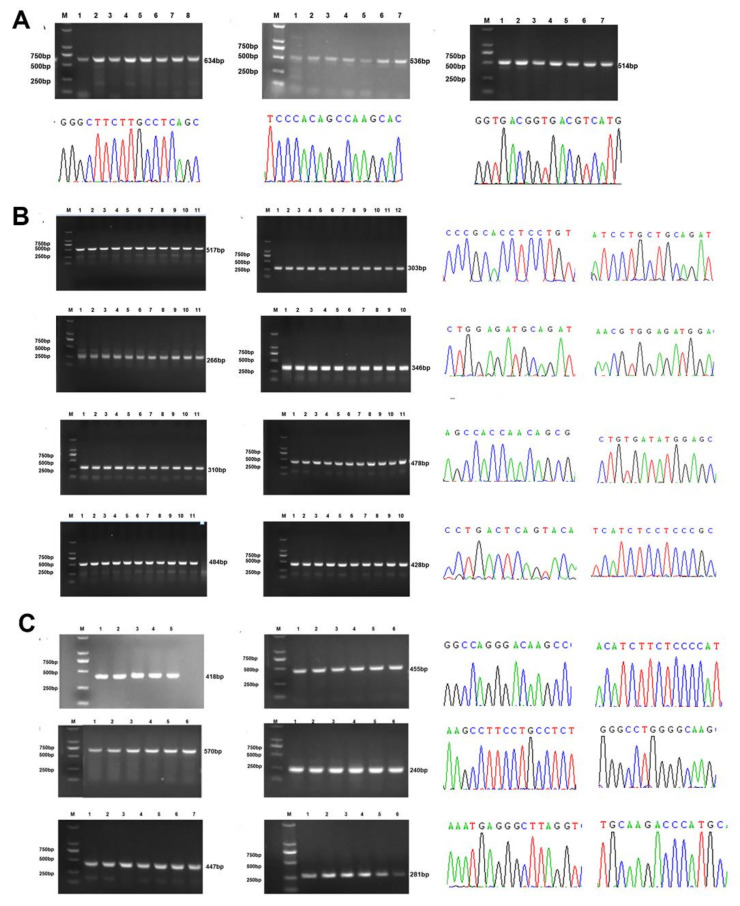
Polymorphism analysis of KRT17 promoter, exons, and introns from 217 Angora rabbits. Detection and sequencing results of KRT17 (**A**) promoter using primers 15–17; (**B**) exons using primers 1–8; and (**C**) introns using primers 9–14 (Table 2).

**Figure 5 ijms-23-06077-f005:**
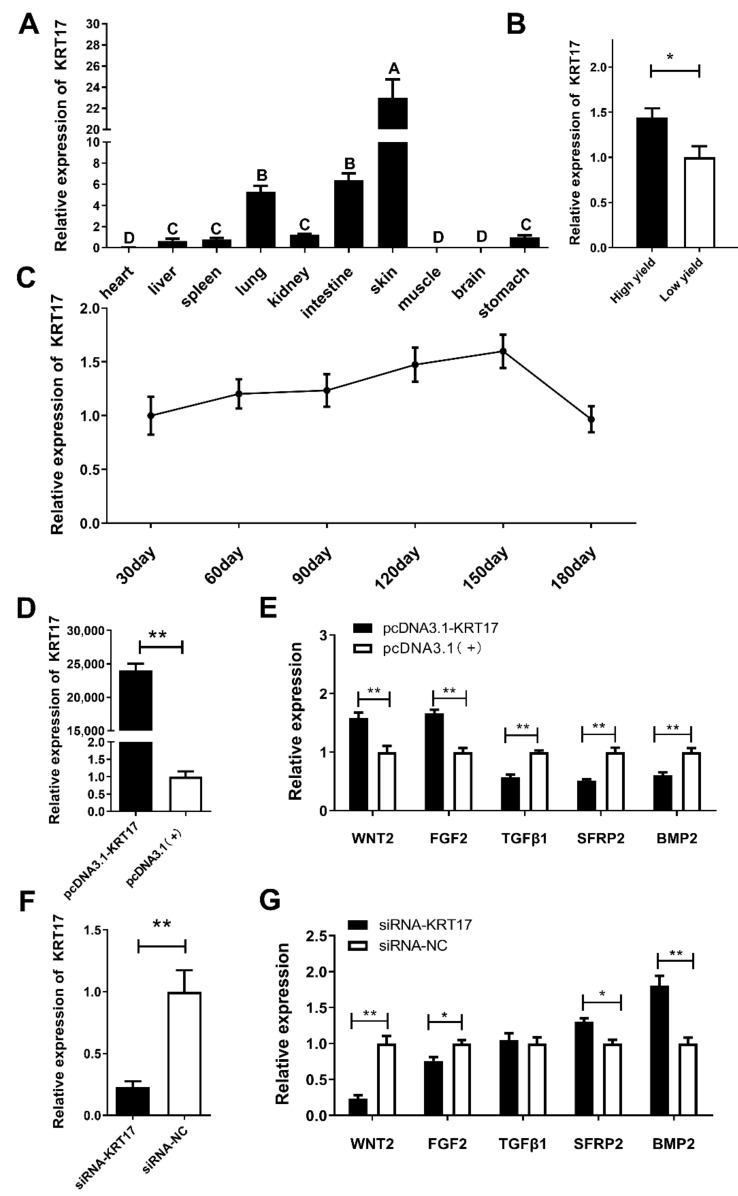
Preliminary analysis of expression and function of KRT17. The expression level of KRT17 in (**A**) different tissues, (**B**) in high and low yield Angora rabbits, and (**C**) in the skin during different hair follicle development cycles; the expression level of KRT17 after (**D**) overexpression and (**F**) RNAi. Expression levels of genes related to hair follicle development after KRT17 (**E**) overexpression and (**G**) RNAi. ** *p* < 0.01. * *p* < 0.05. Different letters indicate significant differences.

**Figure 6 ijms-23-06077-f006:**
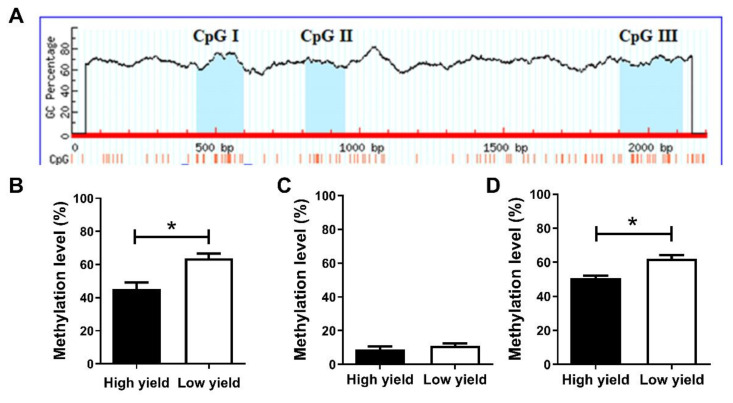
Methylation levels of three CpG islands in the high and low yield groups. (**A**) distribution of CpG islands in rabbit *KRT17* promoter. Methylation levels of (**B**) CpG I; (**C**) CpG II; and (**D**) CpG III islands in the high and low yield groups, respectively. * *p* < 0.05.

**Figure 7 ijms-23-06077-f007:**
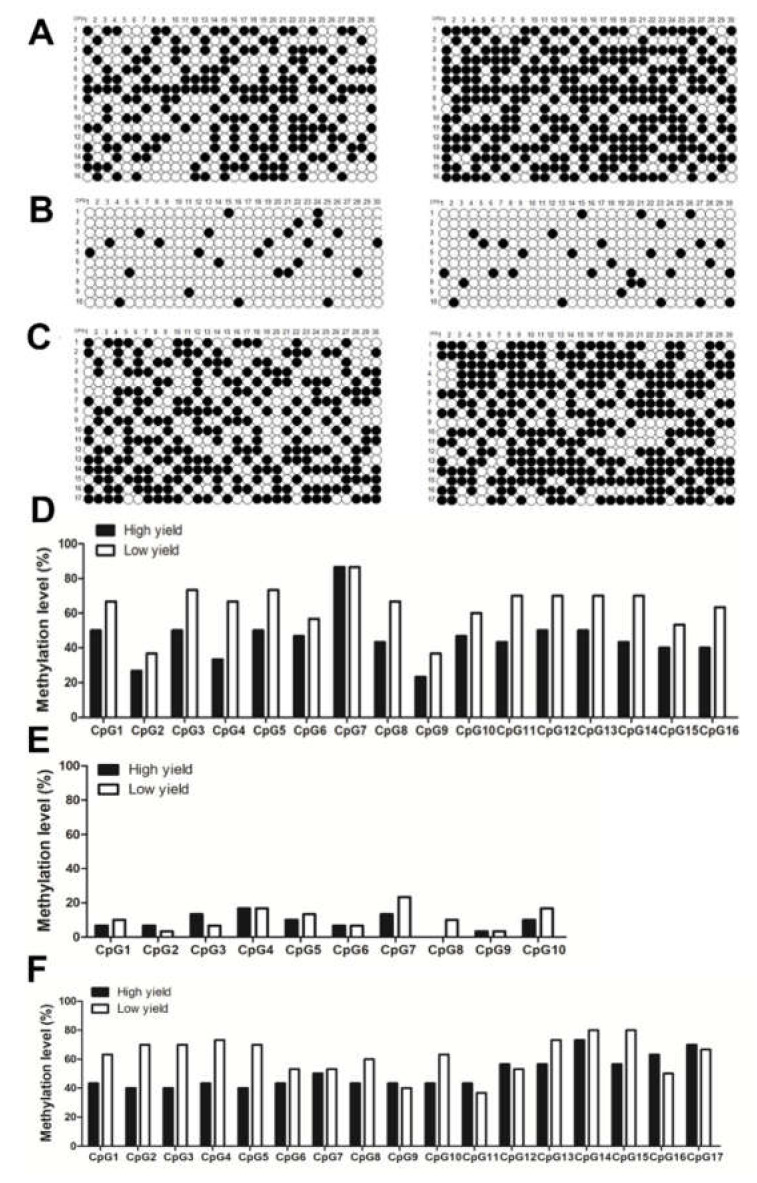
The detection of methylation in high and low yield groups at CpG I–III sites. (**A**–**C**) are the methylation detection patterns of high and low yield groups at CpG I, II, and III sites, respectively. (**D**–**F**) are the methylation level of high and low yield groups at CpG I, II, III sites, respectively.

**Figure 8 ijms-23-06077-f008:**
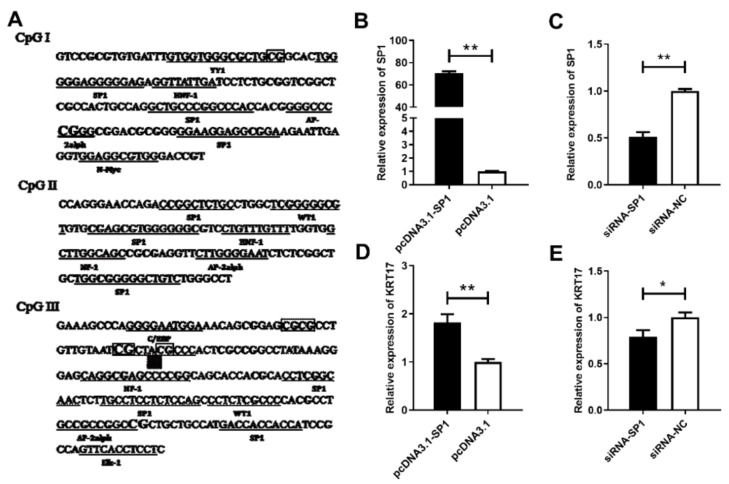
SP1 regulates *KRT17* expression. (**A**) predictive transcription factor binding sites in CpG islands of *KRT17* promoter; (**B**) SP1 overexpression; (**C**) SP1 interference; (**D**) SP1 overexpression; and (**E**) interference effect on *KRT17* expression. ** *p* < 0.01. * *p* < 0.05.

**Figure 9 ijms-23-06077-f009:**
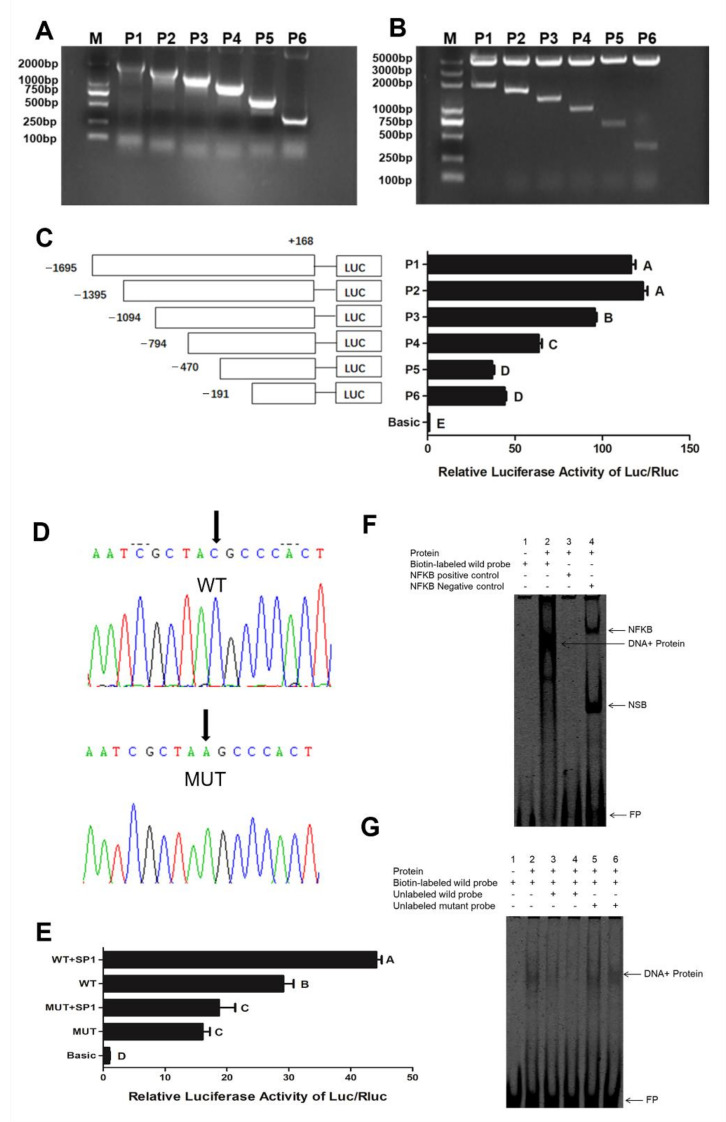
SP1 regulates the activity of the *KRT17* promoter. (**A**) PCR amplification results of different promoter fragments; (**B**) double digestion results of different promoter fragment vectors; (**C**) luciferase activity of the different promoter fragments; (**D**) sequencing results of wild type (WT) and mutant (MUT) plasmids; (**E**) luciferase activity of mutant *KRT17* promoter; (**F**) EMSA assay: Lane 1, empty probe; lane 2, test group; lane 3, negative control NFkB; lane 4, positive control NFkB; (**G**) competitive EMSA. Lanes 1 and 2 are negative and positive controls, lanes 3 and 4 are 33- and 100-fold unlabeled normal probes, and lanes 5 and 6 are 33-fold and 100-fold unlabeled mutant probes for competition reactions, respectively. NFkB was a positive binding protein, DNA + Protein denotes SP1 binding, NSB was the non-specific binding protein, and FP denotes free probe. Different letters indicate significant differences.

**Table 1 ijms-23-06077-t001:** Population weight and wool production data of adult Angora rabbits.

Parameter	Maximum Value	Minimum Value	Average Value	Range
weight (g)	4672	3413	4011.1 ± 216.7	1259
wool production (g)	487.8	228.4	331.1 ± 57.9	259.4

**Table 2 ijms-23-06077-t002:** Weight and wool production data of high- and low-yield groups.

Item	Group	High Yield Group	Low Yield Group
weight (g)	4011.1 ± 216.7	4078.0 ± 200.2	3887.1 ± 190.3
wool production (g)	331.1 ± 57.9 ^b^	443.2 ± 20.8 ^a^	236.2 ± 3.6 ^c^
Quantity	217	20	11

Different letters in the same row indicate significant differences (*p* < 0.05).

**Table 3 ijms-23-06077-t003:** Methylation levels of CpG I sites in the high and low yield groups.

CpG Sites	Group H (%)	Group L (%)	*p*-Value	CpG Sites	Group H (%)	Group L (%)	*p*-Value
CpG1	50.00	66.67	0.295	CpG10	46.67	60.00	0.438
CpG2	26.67	36.67	0.580	CpG11	43.33	70.00	0.067
CpG3	50.00	73.33	0.110	CpG12	50.00	70.00	0.187
CpG4	33.33	66.67	0.019	CpG13	50.00	70.00	0.187
CpG5	50.00	73.33	0.110	CpG14	43.33	70.00	0.067
CpG6	46.67	56.67	0.606	CpG15	40.00	53.33	0.438
CpG7	86.67	86.67	1.000	CpG16	40.00	63.33	0.120
CpG8	43.33	66.67	0.119	**Overall**	**45.20**	**63.75**	**0.000**
CpG9	23.33	36.67	0.399				

H = high yield group. L = low yield group. Bold letters indicate significant sites (*p* < 0.05).

**Table 4 ijms-23-06077-t004:** Methylation levels of CpG II sites in the high and low yield groups.

CpG Sites	Group H (%)	Group L (%)	*p*-Value	CpG Sites	Group H (%)	Group L (%)	*p*-Value
CpG1	6.67	10.00	1.000	CpG7	13.33	23.33	0.506
CpG2	6.67	3.33	1.000	CpG8	0	10.00	0.237
CpG3	13.33	6.67	0.671	CpG9	3.33	3.33	1.000
CpG4	16.67	16.67	1.000	CpG10	10.00	16.67	0.706
CpG5	10.00	13.33	1.000	Overall	9.00	11.00	0.497
CpG6	6.67	6.67	1.000				

H = high yield group. L = low yield group.

**Table 5 ijms-23-06077-t005:** Methylation levels of CpG III sites in the high and low yield groups.

CpG Sites	Group H (%)	Group L (%)	*p*-Value	CpG Sites	Group H (%)	Group L (%)	*p*-Value
CpG1	43.33	63.33	0.195	CpG10	43.33	63.33	0.195
**CpG2**	**40.00**	**70.00**	**0.037**	CpG11	43.33	36.67	0.792
**CpG3**	**40.00**	**70.00**	**0.037**	CpG12	56.67	53.33	1.000
**CpG4**	**43.33**	**73.33**	**0.035**	CpG13	56.67	73.33	0.279
**CpG5**	**40.00**	**70.00**	**0.037**	CpG14	73.33	80.00	0.761
CpG6	43.33	53.33	0.606	CpG15	56.67	80.00	0.095
CpG7	50.00	53.33	1.000	CpG16	63.33	50.00	0.435
CpG8	43.33	60.00	0.301	CpG17	70.00	66.67	1.000
CpG9	43.33	40.00	1.000	**Overall**	**50.00**	**62.16**	**0.000**

H = high yield group. L = low yield group. Bold letters indicate significant sites (*p* < 0.05).

**Table 6 ijms-23-06077-t006:** Correlation coefficient of CpG I methylation rate and KRT17 expression.

CpG Sites	Correlation Coefficient	*p*-Value	CpG Sites	Correlation Coefficient	*p*-Value
CpG1	0.159	0.660	CpG10	−0.670	0.034
CpG2	0.153	0.674	CpG11	−0.417	0.230
CpG3	−0.429	0.216	CpG12	−0.313	0.378
CpG4	−0.201	0.578	CpG13	−0.363	0.302
CpG5	−0.415	0.232	CpG14	−0.566	0.088
CpG6	0.313	0.379	CpG15	−0.134	0.713
CpG7	0.123	0.735	CpG16	−0.192	0.594
CpG8	−0.160	0.659	Overall	−0.319	0.368
CpG9	−0.452	0.189			

**Table 7 ijms-23-06077-t007:** Correlation coefficient of CpG III methylation rate and KRT17 expression.

CpG Sites	Correlation Coefficient	*p*-Value	CpG Sites	Correlation Coefficient	*p*-Value
CpG1	0.109	0.764	CpG10	−0.233	0.518
CpG2	−0.219	0.543	CpG11	0.571	0.085
CpG3	−0.156	0.666	CpG12	−0.055	0.880
CpG4	−0.741	0.014	CpG13	−0.558	0.093
CpG5	−0.389	0.0.266	CpG14	−0.728	0.017
CpG6	−0.202	0.576	CpG15	−0.439	0.204
CpG7	−0.139	0.701	CpG16	0.070	0.847
CpG8	−0.157	0.665	CpG17	−0.280	0.434
CpG9	0.088	0.810	Overall	−0.442	0.200

**Table 8 ijms-23-06077-t008:** RACE primers for KRT17 gene.

Name	Sequence (5′→3′)	Corresponding Experiment
GSP1	CCGACACCCGACAGGA	5′ RACE
GSP2	GGAGGAGCCCTTGATGGA	
GSP3	GAGGAGGTGAACTGGCGG	
C395-1	GAGCAGGAGATCGCCACCTACCGC	3′ RACE
C395-2	TACAAGCCAAAAGAACCCGTGACC	
KRT17-CDS F	ATGACCACCACCATCCGCCAGTT	CDS amplification
KRT17-CDS R	TTAGCGGGTGGTCTGGTGCACCT	

**Table 9 ijms-23-06077-t009:** Polymorphism detection primers to amplify the exons, introns, and promoter regions of the KRT17 gene.

Name	Sequence (5′→3′)	Length (bp)	Annealing Temperature (°C)	Location
KRT17-1	F:CCACGCACCTCGGCAACTCT	517	59.6	Exon
	R:GGCACCAACAAGCCCACCT			
KRT17-2	F:AGGGAGACCACCTCAGAAGCC	303	62.4	
	R:CGTCTGAACCCAGTGACCTTT			
KRT17-3	F:AAGCATCCGGTTCCCACGTC	266	59.6	
	R:CCCTCCCACAGCATCAGCCT			
KRT17-4	F:AGGCTAAACAGGACAAACGAGG	346	59.6	
	R:CTAGGTCCCATCTCCGAGGC			
KRT17-5	F:GTCAGTTTTGGGAGGAGGCT	310	62.4	
	R:GCTCCTTCCCTCTGTGCTTG			
KRT17-6	F:CTGCCTGTGCTGTCTCGTCT	478	55.7	
	R:CCCACTCACTGGGCGTCCTC			
KRT17-7	F:ATGGAGCAGCAGAACCAGGAG	484	55.7	
	R:CCCAGACCCAGGACCACATC			
KRT17-8	F:AAGGCACCAGTGGACATAGGAA	428	62.4	
	R:GGTGTCAGGCAGAGGGAGGT			
KRT17-9	F:CGACTACAGCCACTACTGGAAGA	418	60.8	Intron
	R:GGGCAGTAAGGGATGACGAA			
KRT17-10	F:CTTGGCTTCGTCATCCCTTACT	455	58.8	
	R:CAGGCTTCTGAGGTGGTCTCC			
KRT17-11	F:GCTGACCTGGAGATGCAGATTG	570	59.6	
	R:TGTGGGTGTCTGGGATGAGGA			
KRT17-12	F:CAGAGTGGCAAGAGCGAGAT	240	58.8	
	R:GGGACAGCTGCACGCAGTAG			
KRT17-13	F:ACTCAGTACAAGCCAAAAGAACG	447	60.3	
	R:CCCAGGGCTATGGTCCAATC			
KRT17-14	F:GGGGAAGATTGGACCATAGC	281	57.2	
	R:CCCTCAACCTCACAGCAGACAC			
KRT17-15	F:GGCTTTCTGTCTCCACTTCCTGC	634	58.8	Promoter
	R:AGTCTGCCGATGGCTTCTACCTC			
KRT17-16	F:GAGGTAGAAGCCATCGGCAGACT	536	61.1	
	R:CCAGGTCACAGAGCAGGAAACG			
KRT17-17	F:CCCCGTTTCCTGCTCTGTGA	514	61.1	
	R:GGCGAGTGGGCGTAGCGATT			

**Table 10 ijms-23-06077-t010:** Primers for qPCR.

Name	Sequence (5′→3′)
KRT17	F:AGCTGCTACAGCTTCGGCTCG
	R:CAGGCGGTCGTTGAGGTTCT
WNT2	F:AGCCATCCAGGTCGTCATGAACCAG
	R:TGCACACACGACCTGCTGTACCC
FGF2	F:GTGTGTGCAAACCGTTACCTT
	R:TCGTTTCAGTGCCACATACCAG
TGFβ1	F:CAGGTCCTTGCGGAAGTCAA
	R:CTGGAACGGGCTCAACATCTA
SFRP2	F:ACTTGTGGGTCACGAGCAAA
	R:GTAGTGCTGCGGCTAGAACA
BMP2	F:GACTTCAACAGTGCCACC
	R:TGCTGTAGCCAAATTCGT
SP1	F:CACCATCAGTTCCGCCAGTT
	R:GACGATCCGCTGGTGGTGAA
GAPDH	F:CACCAGGGCTGCTTTTAACTCT
	R:CTTCCCGTTCTCAGCCTTGACC

**Table 11 ijms-23-06077-t011:** Primers for KRT17 and SP1 cloning and RNAi.

Name	Sequence (5′→3′)
KRT17	F:gggagacccaagctggctagcATGACCACCACCATCCGCC
	R:tgctggatatctgcagaattcTTAGCGGGTGGTCTGGTGC
siRNA-KRT17	F:GCAACUACUCCAGCUGCUATT
	R:UAGCAGCUGGAGUAGUUGCTT
SP1	F:gggagacccaagctggctagcATGAGCGACCAAGATCACTCCA
	R:aacgggccctctagactcgagTCAGAAGCCATTGCCACTGAT
siRNA-SP1	F:GGGAAACGCUUCACACGUUTT
	R:AACGUGUGAAGCGUUUCCCTT
siRNA-NC	F:UUCUCCGAACGUGUCACGUTT
	R:ACGUGACACGUUCGGAGAATT

Note: lowercase letters are homologous sequences at the end of the vector.

**Table 12 ijms-23-06077-t012:** BSP primers for rabbit KRT17 promoter.

Methylated Region	Sequence (5′→3′)	Length (bp)
CpG I	F:TTTTTTGGGTTTTTTTTAGTGATTA R:ATTCTTAAAAACCTCCCTAACCTAAC	243
CpG II	F:AGTTTAGGGTTTTGATGAATGATAG R: ATAACTTCTACCTCCCAAACAAAC	201
CpG III	F:GAAAGTTTAGGGGAATGGAAATAG R:AAACCCTTAATAAAACTAAAAAAAATAAAC	216

**Table 13 ijms-23-06077-t013:** Primers of KRT17 promoter for the construction of deletion plasmids.

Name	Sequence (5′→3′)	Length (bp)
KRT17-P1	F:cgagctcttacgcgtgctagcTGCCCCATGTACCCGTCTG	1894
KRT17-P2	F:cgagctcttacgcgtgctagcGCCAGGGAGGTTTCTAAGA	1564
KRT17-P3	F:cgagctcttacgcgtgctagcCTTGGGGAATCTCTCGGCT	1263
KRT17-P4	F:cgagctcttacgcgtgctagcCATCCCACAGCCAAGCACC	963
KRT17-P5	F:cgagctcttacgcgtgctagcTGGCCTGGTGGATTTAGGT	639
KRT17-P6	F:cgagctcttacgcgtgctagcCGCACATGTAGCCGCCCAG	360
KRT17-P	R:acttagatcgcagatctcgagGACAGGAGGTGCGGGACGA	Downstream primer

Note: The lower case letters indicate the restriction sites.

**Table 14 ijms-23-06077-t014:** Primers for C to A mutation in SP1 binding site.

Name	Sequence (5′→3′)	Annealing Temperature (°C)
Mut F	AATCGCTAaGCCCACTCGCCGGCCTATAAAGG	67
Mut R	GAGTGGGCtTAGCGATTACAACAGGCGCGCTC	

Note: lowercase letters indicate the mutated nucleotide.

**Table 15 ijms-23-06077-t015:** Primers for EMSA.

Name	Sequence (5′→3′)
Normal probe	GTTGTAATCGCTACGCCCACTCGCCG
	CGGCGAGTGGGCGTAGCGATTACAAC
Mutation probe	GTTGTAATATATGCGGTCACTCGCCG
	CGGCGAGTGACCGCATATATTACAAC

**Table 16 ijms-23-06077-t016:** EMSA reaction system.

Reagent	Empty Probe	Test Samples	Negative Control	Positive Control
polydI:dC (μL)	0	0.25	0	0
10 × Binding Buffer (μL)	0	1.5	1.5	1.5
protein sample (μL)	0	2	0	0
negative nuclear extract (NFkB)	0	0	0.5	0
positive nuclear extract (NFkB)	0	0	0	0.5
ddH_2_O (μL)	15	11.25	13	13

**Table 17 ijms-23-06077-t017:** Cold competitive EMSA reaction system.

Reagent	1	2	3	4	5	6
10 × Binding Buffer (μL)	0	2	2	2	2	2
PolydI:dC (μL)	0	0.25	0.25	0.25	0.25	0.25
Unlabeled normal probe (μL)	0	0	1.5	3	0	0
Unlabeled mutant probe (μL)	0	0	0	0	1.5	3
Nuclear protein (μL)	0	2	2	2	2	2
ddH_2_O (μL)	20	15.75	14.25	12.75	14.25	12.75
Unlabeled probe: labeled probe ratio			33:1	100:1	33:1	100:1

## Data Availability

All data supporting our findings are included in the manuscript.

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
