# Peer review of "Promoter Methylation Changes in KRT17: A Novel Epigenetic Marker for Wool Production in Angora Rabbit"

_ijms, 2022, doi:10.3390/ijms23116077_

Round 1

Reviewer 1 Report

The manuscript describes the analysis of the KRT17 gene. SNP sites were not found. Among the methylation sites, some showed association with wool quantity and expression level of KRT17. ’CpG III site 4 showed a significant correlation with both wool production and KRT17 expression’.

After minor modification, I suggest the MS for publication.

Notifications, suggestions:

Line 22

‘The methylation level of the KRT17 promoter was determined by single nucleotide polymorphisms.’

What does that mean? Please rephrase!

Line 19

 ’The full-length sequence of KRT17 was obtained by rapid amplification of 18 cDNA ends technology, and the SNPs were detected in the promoter, exon, and intron regions by direct sequencing.’

The sentence above contradicts the whole MS and, for example, the following sentences as well:

Line 25: ’..map of KRT17 was drawn, and no SNPs were found in the promoter, exon, and intron…’

line 96: ’ Also, no mutation sites were found…’

Line 29

 ’…significant associated with low wool…’ 

is to be changed to

’…significant association with low wool…’ 

Line 55

’…overall or related gene methylation...’

Related to what?

Line 75

’… The effective production data of 217 Angora…’

is to be changed to

‘…The production data of 217 Angora…’

Line 83

’…was not affected by weight change meeting the screening requirements.’

is to be changed to

‘…was not affected by weight change meeting.’

Line 197

Figure 8 might be not necessary to show.

Line 257

…KRT17 is relatively conserved in the population of Angora rabbits showing no SNPs.’

is to be changed to

‘…KRT17 is conserved in the population of Angora rabbits showing no SNPs.’

Author Response

Dear Editors and Reviewers:

Thank you for your letter and for the reviewers’ comments concerning our manuscript entitled “Promoter methylation changes in KRT17: a novel epigenetic marker for wool production in Angora rabbit” (ID: 1688183). Those comments are all valuable and very helpful for revising and improving our paper, as well as the important guiding significance to our researches. We have studied comments carefully and have made correction which we hope meet with approval. Revised portion are marked up using the“Track Changes” in the paper. The main corrections in the paper and the responds to the reviewer’s comments are as flowing:

Responds to the reviewer’s comments:

The manuscript describes the analysis of the KRT17 gene. SNP sites were not found. Among the methylation sites, some showed association with wool quantity and expression level of KRT17. ’CpG III site 4 showed a significant correlation with both wool production and KRT17 expression’.

After minor modification, I suggest the MS for publication.

Notifications, suggestions:

Line 22

‘The methylation level of the KRT17 promoter was determined by single nucleotide polymorphisms.’What does that mean? Please rephrase!

Response: We are very sorry for our incorrect writing. I have revised it.

‘The methylation level of the KRT17 promoter was determined by Bisulfite Sequencing PCR.’

Line 19

 ’The full-length sequence of KRT17 was obtained by rapid amplification of 18 cDNA ends technology, and the SNPs were detected in the promoter, exon, and intron regions by direct sequencing.’ The sentence above contradicts the whole MS and, for example, the following sentences as well:

Line 25: ’..map of KRT17 was drawn, and no SNPs were found in the promoter, exon, and intron…’

line 96: ’ Also, no mutation sites were found…’

Response: We have re-written this part according to the Reviewer’s suggestion. The statements of ‘the SNPs were detected’ were corrected as ‘the polymorphism was analyzed’ .

Line 29

 ’…significant associated with low wool…’ 

is to be changed to

’…significant association with low wool…’ 

Response: We have made correction according to the Reviewer’s comments.

Line 55

’…overall or related gene methylation...’

Related to what?

Response: Thanks to you for your good comments. We have re-written this part ‘the whole-genome DNA or related gene methylation level’.

Line 75

’… The effective production data of 217 Angora…’

is to be changed to

‘…The production data of 217 Angora…’

Response: We have made correction according to the Reviewer’s comments.

Line 83

’…was not affected by weight change meeting the screening requirements.’

is to be changed to

‘…was not affected by weight change meeting.’

 Response: We have made correction according to the Reviewer’s comments.

Line 197

Figure 8 might be not necessary to show.

Response: Thanks to you for your good comments. We have deleted Figure 8.

Line 257

 …KRT17 is relatively conserved in the population of Angora rabbits showing no SNPs.’

is to be changed to

‘…KRT17 is conserved in the population of Angora rabbits showing no SNPs.’

Response: We have made correction according to the Reviewer’s comments.

Reviewer 2 Report

Dear authors,

This manuscript presents an interesting research work regarding molecular markers associated with woll production of angora rabbits.

The paper is well organized, well argued, the methodology is presented in a clear way and the language of the text allows the target-readers to fully comprehend the concept of the study, the methods and its results.

Consequently, this paper should be accepted for publication.

Author Response

Dear Editors and Reviewers:

Thank you very much for your efforts on our paper. On behalf of my co-authors, we would like to express our great appreciation to editor and reviewers.